# Study of the Possibility of Using Virtual Reality Application in Rehabilitation among Elderly Post-Stroke Patients

**DOI:** 10.3390/s24092745

**Published:** 2024-04-25

**Authors:** Katarzyna Matys-Popielska, Krzysztof Popielski, Anna Sibilska-Mroziewicz

**Affiliations:** 1Institute of Metrology and Biomedical Engineering, Warsaw University of Technology, 02-525 Warsaw, Poland; krzysztof.popielski.dokt@pw.edu.pl; 2Institute of Micromechanics and Photonics, Warsaw University of Technology, 02-525 Warsaw, Poland; anna.mroziewicz@pw.edu.pl

**Keywords:** virtual reality, rehabilitation, elderly patients, stroke, paresis, unilateral spatial neglect

## Abstract

Thanks to medical advances, life expectancy is increasing. With it comes an increased incidence of diseases, of which age is a risk factor. Stroke is among these diseases, and is one of the causes of long-term disability. The opportunity to treat these patients is via rehabilitation. A promising new technology that can enhance rehabilitation is virtual reality (VR). However, this technology is not widely used by elderly patients, and, moreover, the elderly often do not use modern technology at all. It therefore becomes a legitimate question whether elderly people will be able to use virtual reality in rehabilitation. This article presents a rehabilitation application dedicated to patients with upper limb paresis and unilateral spatial neglect (USN). The application was tested on a group of 60 individuals including 30 post-stroke patients with an average age of 72.83 years. The results of the conducted study include a self-assessment by the patients, the physiotherapist’s evaluation, as well as the patients’ performance of the exercise in VR. The study showed that elderly post-stroke patients are able to use virtual reality applications, but the ability to correctly and fully perform an exercise in VR depends on several factors. One of them is the ability to make logical contact (*p* = 0.0001 < 0.05). However, the study presented here shows that the ability to use VR applications does not depend on age but on mental and physical condition, which gives hope that virtual reality applications can be used in post-stroke rehabilitation among patients of all ages.

## 1. Introduction

The world is constantly developing and moving forward. There are constant advances in medicine, which, among other things, have been seen over the years through the invention and spread of antibiotics, vaccines, and other drugs, the development of surgical techniques and emergency medicine, or advances in disease diagnosis and prevention. Living conditions are also improving through access to clean water, food and sanitation, health care, or increased affluence allowing for better nutrition, among other things. Lifestyles are also changing with increased awareness of healthy eating, physical activity, and the harm of smoking and alcohol abuse. All of this contributes to an increase in life expectancy. According to WHO data, life expectancy at birth worldwide has increased from 46.5 years in 1950 to about 73.0 years in 2019. Life expectancy is steadily increasing [1]. Despite the decline in life expectancy caused by the COVID-19 pandemic, it is estimated that it will reach 77.0 years in 2048 [2].

Increasing life expectancy also has negative consequences. Age is a risk factor for many diseases such as stroke, Alzheimer’s disease, and Parkinson’s disease, as well as many cancers. The incidence of stroke increased by 70.0% between 1990 and 2019 in the U.S., and the prevalence increases with age [3]. Due to the increase in the total number of stroke cases, as well as the decrease in stroke-related mortality, age is an important aspect of the occurrence of stroke, as well as other diseases, and the complications they bring with them. Stroke is one of the leading causes of long-term disability [4]. Complications of stroke involve the cardiovascular, respiratory, gastrointestinal, musculoskeletal, or neurological systems, among others. Complications include upper limb paresis, which is the most common complication after stroke [5], or neglect syndrome, which affects about 25–30% of patients [6]. Such post-stroke complications, the occurrence of neurodegenerative diseases such as Parkinson’s disease, or other diseases that limit, among other things, the dexterity of the upper limbs, and thus the ability to function independently without the help of others, also make important the social aspect of these diseases.

The chance to improve upper limb motor function and normal functioning for these patients is rehabilitation [7], and the increase in the incidence of these conditions is creating a greater demand for rehabilitation services [8]. Conventional rehabilitation of the upper limb usually requires systematic daily training for 6 consecutive weeks for about 2–3 h a day [9]. Due to the intensity of rehabilitation, it is expensive and resource-intensive and is not always widely available [10]. In addition, due to the duration and lack of widespread access to specialized equipment, rehabilitation can be monotonous and exhausting for patients and is also burdensome for physiotherapists [11]. Due to these limitations, new technologies have begun to be used in upper limb rehabilitation. Due to its potential, one of the most widely used technologies has become both non-immersive and immersive virtual reality (VR) [12].

The difference between non-immersive and immersive virtual reality is the degree of immersion in an artificial environment. A non-immersive VR system usually consists of a standard computer monitor with a mouse, remote control, or joystick. Such a system is limited in its ability to produce immersion, usually lacking the ability for the image to change according to the movement of the body and head in real time [13]. It is different from immersive virtual reality, where a head-mounted display (HMD) headset is used. It allows the display of a virtual, three-dimensional world and enables interaction using one’s own body, giving the impression of being completely transported into another environment. The total immersion into another reality allows for a more tangible experience than with other video formats [14]. A person immersed in a virtual world experiences multi-sensory stimulation and, through a sense of presence in the virtual world, responds naturally, with full involvement, as if they were part of the virtual world [15]. Moreover, due to the transfer of rehabilitation to the virtual world, engaging multiple senses, and allowing patients to gamify, they show increased motivation and enthusiasm compared to exercises performed during conventional rehabilitation [16,17]. Virtual reality also makes it possible to simulate activities of daily living, so that patients can experience self-care training in an engaging way [18]. It also has the advantage of allowing rehabilitation without the physiotherapist’s active participation throughout the rehabilitation [19]. The physiotherapist’s tasks are provided by a virtual environment. This reduces the workload of physiotherapists, thereby increasing access to rehabilitation.

Thanks to the development of technology and its increased availability, the use of virtual reality in rehabilitation is becoming more common. Among the many studies, it is possible to find papers summarizing current research. One of these papers [20] reports that out of 152 studies reviewed, 54% relate to neurological rehabilitation, 14% to orthopedic rehabilitation, and 8% each to burn and gerontological rehabilitation. These studies are mainly concerned with the effectiveness of patient treatment and motor development. Considering the age of the patients, respectively, 30% and 27% of those participating in the studies were over 60 and in the 45–59 age group. However, it should be noted that most of the studies used video games on commercially available consoles such as the Nintendo Wii, Xbox Kinect, and PlayStation EyeToy (58%), and only 2% of the studies dealt with immersive VR.

Existing studies on the use of virtual reality in the rehabilitation of post-stroke patients focus on several aspects: motor recovery (improving balance, lower limb function, upper limb function, muscle strength, gait speed) [21,22], sensory recovery (reducing speech impairment, enlarging visual field) [23], and cognitive recovery (improving the ability to plan, initiate independent actions, understand, solve problems) [24]. The results of these studies show that virtual reality is a promising and effective form of rehabilitation. It can be used at different stages of disease, with the greatest effectiveness observed in motor rehabilitation, where patients show improvement in motor functions. As a result, the patient gains independence, which improves their quality of life and reduces the burden on caregivers [25]. Also, the review paper [20] confirms the advantages of VR in the rehabilitation of various conditions such as improving balance and gait, increasing motor skills and mobility, improving exercise performance, enhancing motor functionality, and improving quality of life. At the same time, it points out the main barriers to the implementation of VR in rehabilitation, such as the high cost of technology, low availability of suitable games for rehabilitation, technical limitations of the device, and lack of standardization in the way of performing exercises.

Existing studies can also be divided according to the technology used. A comparison of studies using non-immersive VR for rehabilitation [13,26,27] versus studies using immersive VR [28,29,30] shows that with immersive VR, the patient experiences a more realistic experience due to being fully immersed in this environment. The findings suggest that some problems such as technical limitations of the device can be solved by using immersive VR and that it is more effective to use immersive VR, as it affects the patient’s involvement in the therapy, which can enhance the effects of motor training [29,31].

A potential problem in using virtual reality in neurological rehabilitation is the ability to use this technology. The 2019 statistics for Poland’s population state that in the group of people aged 65+, 92.3% use a TV, but only 25.9% use an internet-connected device. Moreover, 78.2% of people in this age group use a cell phone, with the majority being a key telephone. Even fewer, only 13.8% of people over 65, use a laptop computer [32]. Internet is used regularly by 33.3% of people over 65 years of age [33]. Moreover, data [34] show that by 2022, only 4.4% of Poles and 8.6% of U.S. residents had ever used VR goggles, but the same data show that the number of VR goggles users is steadily increasing. Thus, in 2023 already, 6% of Poles and 9.8% of U.S. residents have ever used VR googles. Despite the growing potential of this technology, given the prevalence of stroke in different age groups and the fact that most stroke patients are elderly, it can be concluded that patients have little experience with the technology. Therefore, it is uncertain whether they will be able to use VR applications.

Previous studies of virtual reality applications for rehabilitation have been based mainly on testing the effectiveness of the games themselves. The average age in many studies [17,29,35] in the experimental groups are between 46 and 59 years old. Thus, they did not explore the use of virtual reality applications among older patients.

Given the increasing trend of using virtual reality in rehabilitation, the age of post-stroke patients, the fact that the vast majority of studies focus on using technology other than immersive VR, and the low level of HMD use especially among the elderly, it becomes reasonable to investigate the feasibility of using immersive virtual reality for the rehabilitation of the upper limb among elderly people (over 60) after a stroke, which was the purpose of this study.

## 2. Materials and Methods

An application was developed to explore the possibility of rehabilitating the elderly using a virtual reality application in the Unity engine (version: 2021.3.19f1, Unity Technologies, San Francisco, CA, USA) with the Unity XR Toolkit toolbox. The application is dedicated to the Meta Quest 2 device Meta, Menlo Park, CA, USA. The set used for rehabilitation exercises uses the HMD Quest 2 device and the included controllers. The HMD technology together with the controllers is sufficient for the presented application, i.e., it does not require the use of additional accessories such as headphones, keyboards, or additional sensors. The application was designed to reflect the rehabilitation of post-stroke patients, particularly patients with upper limb paresis and unilateral spatial neglect.

### 2.1. Conventional Rehabilitation

Post-stroke rehabilitation enables patients to gain new skills in the field of self-care during the first year after a stroke and efficiently limits the loss of such abilities over time [36]. The rehabilitation scope is adapted to the patient’s needs. Training covers deteriorated functions. It is based on effective learning principles, using brain plasticity. The objective of physiotherapy in a patient with a motor deficit, both resulting from the primary motor cortex as well as being more complex, is recreating motor abilities or their compensation. The motor deficit may be mitigated directly through treatment with movement under the active participation of the patient. At the same time, it is important that the extent of rehabilitation can be adjusted during rehabilitation according to the patient’s needs and progress.

In the event of a neglect syndrome, the exercises should involve various teleceptive, auditory, and visual stimuli that should direct the patient’s attention to the neglected side. It is important for both the exercises of limb on the neglected side and the training of visual and spatial searching to be adapted to the change or, if possible, corrected patient centerline. Such exercises are mainly based on visual searching on the non-neglected side in order to find an object to grab, and on the neglected side, to place the object in the target space. The motion of the eyes or limb itself should be conducted gradually, starting from the non-neglected side and towards the neglected side (until a line or defined point/object clearly seen by the patient). Rehabilitation exercises similar to exercises in paresis are also aimed at training object grabbing and manipulating, within both the neglected and non-neglected sides [37].

### 2.2. VR Application for Rehabilitation

The app is designed to provide the most pleasant experience for the patient while keeping the environment realistic and meeting the basic tenets of rehabilitation for post-stroke patients with hemiparesis or USN (grasping training, object manipulation, implementation of exercises on both sides of the body, and the ability to adjust the difficulty of the exercise to the patient’s ability). In addition, two assumptions based on the age and condition of rehabilitated patients are important. Rehabilitation should take place in a sitting position, while the application itself should be as straightforward as possible, allowing most of the attention to be focused on the movement used in rehabilitation.

The virtual environment is shown in Figure 1. It is a room where the patient sits on a couch. A small table is placed in front of him, and typical furniture is placed against the walls. Adding to the typical environment is a glass wall and ceiling with a view of the forest. This setting keeps the environment real and provides pleasant, relaxing views.

A diagram of the game is shown in Figure 2. The first step in the game is to choose a side (Figure 1). The choice of side is important in patients with USN, as it determines the direction of the exercise. At the same time, this choice is essentially intuitive for these patients, as they have an unequal range of visual fields on the right and left sides. They see more from one side than the other. Thus, they select the element on the side they can see. In contrast, side selection is less important in patients with paresis. It only determines the direction of the exercise while the mechanics of the game remain the same. Therefore, rehabilitation exercises for these users will involve the same activities of grasping and manipulating objects.

After starting the game and selecting a side, a red apple appears on the table on the selected side. On the other side, a white target box with a green border appears on the table. On the selected side, buttons appear under the table: reset (white) and level buttons (green, yellow, red). Also on this side, information about the number of points collected appears on the glass wall. The arrangement of the elements is shown in Figure 3. The placement of elements is dictated by the patient’s ability to see the space. As a result, a patient who neglects part of the space will be able to use the level buttons and the reset button without any problems, since they are located in the space that this person perceives.

The objective of the game is to transfer as many apples to the target field. The fruit can be grabbed with both the right and left hand, which makes the game universal since it can be operated with one or both upper limbs.

The course of the exercise is as follows: reaching out with the hand holding the controller towards the apple, grabbing the apple-pressing a button on the controller (Figure 4), analogous to clenching one’s fingers on the apple, moving the apple to the target field, dropping the apple in the target field by opening your hand and releasing the buttons as if you were dropping an object. Each time an apple is correctly placed in the target location, the appropriate number of points is added to the total score, and the new apple and target field are placed in their respective positions on the table, keeping the side selected in the first step.

In addition to locating the apple on the non-neglected side, other elements have been implemented in the game to help the patient perform the exercise. In order to help the patient locate the field, it is surrounded by a dashed line leading towards the apple. In addition, arrows appear above the table that go from the selected side to the other side. In this way, they indicate the direction of the target field and direct the patient’s attention to the skipped side. Moreover, to help both the patient and the physiotherapist, haptic responses have been implemented when the fruit is correctly grabbed as well as put down on the required field. In this way, the patient gets an additional stimulus for the correct execution of the exercise, while the physiotherapist gets an additional opportunity to control or explain the exercise.

The game allows the therapy range to be adjusted to the patient’s range of motion by implementing three levels. At any time, the user can decide to change the level by tapping the appropriate level button. Each level differs in terms of apple and target field location, or rather the ranges/area where they can appear. The exact location of the apple and the target field is chosen randomly from the range assigned to each level. The harder the level, the location of the apple and target field will be farther from the central line. In addition, each level differs in the number of points that are awarded for the correct execution of the exercise. In case a user is unable to reach the apple or the target field on the table, the user can tap the reset button, which will cause a new apple and target field to appear on the table according to the current level.

In addition, after 5 subsequent points are exceeded, a random “motivator” is displayed on the selected side-Figure 5, on the left. It is a ball with the symbol “$” or “€” or with a motivating phrase, such as “Fantastic” or “Go on!”. The aim of the “Motivators” is to partially reflect a physiotherapist, who tells a patient who is being rehabilitated that an exercise is being correctly done or motivates the patient to continue exercising. Besides the points displayed on the right or left side, the game also offers other statistics. They are a good indicator of the patient’s rehabilitation progress. They show both the general rehabilitation progress, limb capacity increase (exercise duration), and patient’s vision range. Statistics are shown in Figure 5. They are displayed in Polish because this is the language of the patients.

### 2.3. Participants

The study included 60 subjects, who were divided into control and study groups, with the control group divided into three (Table 1). The first group was 10 young, healthy subjects (3 women, 7 men) with a mean age of 21.3 years (SD 0.48, range: 21–22). The second group was 10 middle-aged people (7 women, 3 men) without neurological conditions—mean age 53.1 years (SD 7.26, range: 39–59). The third group was 10 elderly people (4 women, 6 men) without neurological conditions—mean age 76.5 years (SD 8.53, range: 63–87). The fourth group was elderly post-stroke patients (11 women, 19 men) with a mean age of 72.83 years (SD 8.14, range: 60–89). All patients in the fourth group had suffered an ischemic stroke. For the first three groups, the criteria for inclusion in the study were age in the range of 20–25, 35–59, and over 60 for groups one, two, and three, respectively; the absence of neurological conditions and other conditions that may affect visual field and upper limb function; and consent to participate in the study. For group four, criteria included age over 60, experience of stroke, consent to participate in the study, and qualification by a physiotherapist based on the absence of contraindications to the use of the VR headset. Among the post-stroke patients, 19 had upper limb paresis and 5 had side skipping syndrome. A total of 11 patients had experienced a stroke in less than 10 days before the study, and 19 in more than 10 days preceding the study. In the first and second groups of subjects, 4 and 3, respectively, had previously used a VR headset. None of the study participants in the third and fourth groups had previous experience with a VR headset. Ethical approval for this trial was obtained from the Warsaw University of Technology Ethics of Research with Human Subjects Team (approval numbered 1/2023).

## 3. Results and Discussion

The results of the study in all groups consist of two parts: the subject’s self-assessment and the performance of the exercise in the application. In addition, patients in group four were additionally evaluated by a physiotherapist. The subjective part of the evaluation was the participants’ self-assessment of whether they would like to try it without suggestion/request and whether they were convinced to try the new technology in exercise/rehabilitation, and after the study, they evaluated whether they were satisfied with this form of exercise. This way of assessing people’s experiences, feelings, or emotions is often used in studies such as a study of the impact of a training program on social isolation, loneliness, and attitudes toward technology in older adults [38], a study of VR applications to assess upper limb motor impairment [39], in investigating the feasibility of using commercial HMD equipment in people with Parkinson’s disease [40] and an assessment of seniors’ perceptions and attitudes toward a VR 360-degree application [41]. The objective aspect of the app’s evaluation was exercise performance. This was divided into two components: the performance of the first part of the therapy—whether the participant was able to grasp the apple in VR—and the performance of the entire therapy—whether the participant was later able to put it down on the target field. In the case of the fourth group, the subjective but experience-based evaluation was the physiotherapists’ assessment of whether the patient was in good or poor contact (good logical contact, in the sense of the study being conducted, can be defined as the ability to communicate seamlessly with the participant, to understand instructions and explanations from the physiotherapists) before the test, and after the test, they assessed whether the patient moved during the rehabilitation using virtual reality in such a way that the rehabilitation premise could be considered fulfilled. The evaluation of each element was done in a zero–one manner. A summary of the results is shown in the Table 2 and Table 3.

According to data [32,33], as age increases, the use of the technology decreases, which the study also showed, as none of the patients over the age of 60 had any previous exposure to virtual reality. Therefore, it becomes a justifiable question whether patients were convinced to try VR technology in rehabilitation. Therefore, within the control groups, the application was first tested on young people, then on middle-aged people, then with elderly people without neurological conditions, and only then on a study group of elderly patients after stroke.

In the group of young people, everyone wanted to try using the VR application. Additionally, 100% of the individuals in this group managed to complete the entire exercise. All respondents were satisfied with the game, they indicated that the game was enjoyable, but too simple for people without additional problems. In the middle-aged group, 90% of the participants wanted to try the VR application. One individual managed only half of the task, while the others completed the entire exercise. Nine people were satisfied with using the app. Among older people without neurological conditions, only half of the respondents were interested in the VR app from the beginning and wanted to try it without persuasion. Overall, 90% of the participants in this group managed to do the first part of the exercise, and 80% managed to do the entire exercise. After the test, 9 out of 10 people in this group positively evaluated the application. The results from the three groups show that there is a statistically significant difference between the willingness to try virtual reality apps by age group (*p* = 0.0095 < 0.05) with older people being less willing to try VR applications than younger people. At the same time, there is no statistically significant difference between the ability to perform part or all of the exercise as well as satisfaction with the app after testing it (*p* equal to 0.381, 0.354, 0.612, respectively).

A challenge in implementing VR technology in the rehabilitation of the elderly encountered during the study was convincing and encouraging respondents to try using the app. For the study group of people over 60 years old after a stroke, 47% of patients wanted to try VR technology in rehabilitation, with 61% in the 60–75 group and only 25% in the 75–90 group. Patients said that this technology was not for them, and that they were not suitable for such therapy. This confirms the supposition that older patients are more skeptical about the study. This is due to the fact that many of them do not have everyday contact with technology such as smartphones/tablets/laptops [32,33]. However, once this barrier is broken and the app is used for the first time, this problem does not occur. This indicates the need for physiotherapist to introduce rehabilitation apps. Despite initial uncertainty, after performing virtual reality rehabilitation, 75% of patients were satisfied with virtual reality therapy and felt they could rehabilitate in this way. Similar results, but using a different technology, were obtained during the AGE-ON program [38], which assisted elderly people in using a tablet. Attitudes toward technology and frequency of use improved during the program, confirming the possibility of using modern technology among elderly patients.

In the fourth group, when performing virtual reality rehabilitation, not all patients were able to perform the exercise. Among all patients, 80% managed to perform the first stage of the exercise and 57% the entire exercise. There may be several reasons for this fact. First, the patients were in different mental states, with different capacities for logical contact. All patients in good contact were able to perform the first stage of the exercise, and 95% of them were able to perform the entire exercise correctly. In the case of patients in poor contact (who have difficulty understanding commands, and thus also have difficulty performing the exercise with traditional rehabilitation), only 25% of patients were able to perform the first stage of the exercise, and no one was able to perform the entire exercise. The ability to perform an exercise in VR strongly depends on the level of patient contact (*p* = 0.0001 < 0.05). Second, the physical condition of the patients varied. Just as with traditional exercises, not all patients are able to perform every exercise, the same is true for exercises in VR. Patients without additional complications, such as hemiparesis or neglect syndrome, perform better with exercises in VR relative to patients with these complications (*p* = 0.0151 < 0.05). Third, in the group of post-stroke patients, the ability to perform the entire exercise is dependent on the time since the event (*p* = 0.0123 < 0.05). This is consistent with the psycho-physical state of the post-stroke patient, as this improves over time. Moreover, in addition to objective and measurable reasons, the ability to perform an exercise in VR may depend on personal ability, which varies with age, experience, or cognitive skills, which depend on many factors, including the level of education [42]. However, there was no correlation between patient age and the ability to perform the exercise (*p* = 0.362 > 0.05), so there is no reason to suspect that age is a barrier to rehabilitation using VR.

In addition, and very importantly, there was no correlation between the ability to perform the exercise between study group four and control group three (*p* = 0.196 > 0.05), which may suggest that the occurrence of a stroke itself does not affect the ability to perform the exercise, but is influenced by the patient’s condition and the presence of additional complications as presented above. Thus, the feasibility of exercise in the app is analogous to that of traditional therapy, in which exercises that are simple for a healthy person are demanding for a person suffering from paresis or neglect syndrome.

Just as important as the ability to perform the exercise is whether the patient was moving. Or, more specifically, whether during the VR-based rehabilitation exercise attempt, they were moving in such a manner that the rehabilitation premise can be considered fulfilled. Among all patients, during the VR exercise, 87.5% of patients moved in such a way that they actively performed the movement expected of a patient in rehabilitation. A correlation was observed between the patients’ self-assessment of their satisfaction with the use of VR and the objective assessment conducted by the physical therapists (*p* = 0.0343 < 0.05). This may indicate that the subject’s interest in the exercise performed increases their motivation and thus improves their performance. In addition, there was no correlation between the ability to perform the exercise correctly and completely, and performing an active movement that can be considered rehabilitation (*p* = 0.235 > 0.05). So, the patient does not need to do the entire exercise to rehabilitate. This is because the essence and basis of rehabilitation is movement, not the performance of a specific exercise [36,43]. This is important, especially among patients in the early stages, who also have difficulty with even basic exercises. Among such patients, it is important to motivate them to continue therapy despite small movements. In this context, another advantage is that during rehabilitation using VR, the patient does not see their actual posture, so the patient’s senses can be deceived. This is because body perception is flexible, changing under the influence of stimuli such as virtual reality [44]. In the course of the study, in several cases, physiotherapists found that patients moved to a greater extent during VR exercises than in traditional rehabilitation. The explanation for this is precisely the change in perception, resulting in a lack of awareness of their limitations.

The fact that patients are moving while performing exercises in virtual reality is promising and may indicate the possibility of using the presented game in rehabilitation. It meets the basic premise of the rehabilitation of post-stroke patients with upper limb paresis—patients perform active movement. The essence of movement also makes it meet the premise for rehabilitation of post-stroke patients with unilateral neglect syndrome—training to connect the two sides of space. Promising in the use of the presented application in rehabilitation is the fact of the effectiveness of other VR applications in post-stroke rehabilitation. Studies [22,45] have shown that virtual reality reflection therapy along with conventional therapy may be more beneficial in improving lower limb function. Many works [21,46,47] indicate that upper limb motor tasks in VR facilitate motor recovery. Studies [26,30] independently show that virtual reality training improves spatial attention and spatial attention transfer to the neglected side. Despite promising results from other studies, further research would need to be conducted to confirm the feasibility of using the presented application in rehabilitation. They should focus on increasing the sample size and diversifying patients’ conditions to evaluate the broader use of VR in rehabilitation. The next step in the research should be to analyze the long-term effects of VR-based rehabilitation. Then, the results of patients using VR for rehabilitation should be compared with a control-group-only traditional rehabilitation method.

In addition to the benefits of improving impaired motor skills, the use of virtual reality in rehabilitation reduces depressive mood in post-stroke patients [24]. Moreover, as mentioned earlier, 75% of patients were interested in rehabilitation using VR. They considered this form of rehabilitation to be more interesting and which may affect their motivation to exercise. Similar results were obtained in a study [23], where positive attitudes and interest in rehabilitation using VR were observed. In addition, studies that relied on self-assessment surveys show that research about VR evidences the potential of improving senior citizens’ well-being by promoting socially engaged states [41].

The studies described above are supported by the literature [20], because just as the use of virtual reality in rehabilitation has been thoroughly described, there is still little knowledge about its applicability in post-stroke rehabilitation for elderly patients. The results regarding motivation and the prevalence of immersive systems for rehabilitation are confirmed in the literature [19,48]. Other researchers [48,49] point out the danger of cybersickness, the equivalent of motion sickness for VR technology. In the described study no such case was found; however, it is necessary to analyze this risk as well as apply methods to compensate for this phenomenon [28] for later long-term studies.

## 4. Conclusions

The article presents a study of the possibility of using VR games in the rehabilitation of the upper limb among elderly post-stroke patients. For this purpose, a VR application dedicated to patients with motor deficits in the upper limb was developed. Studies have confirmed that people over 60 years of age in their first contact with virtual reality are not convinced to use it. At the same time, these individuals, after using a dedicated rehabilitation application, most likely change their minds and are satisfied with the VR therapy session. Moreover, they are interested in further therapy in this form. In addition, the study found that the ability to perform a rehabilitation exercise in VR depends primarily on the patient’s condition, both motor skills and mental condition. The ability to perform an exercise in VR, on the other hand, does not depend on age. The majority of patients are able to perform rehabilitation exercises in VR, to the same extent as exercises performed during conventional therapy. Virtual reality offers many possibilities and is a promising tool in rehabilitation. The study gives hope that it can be used by people of all ages, including the elderly.

## Figures and Tables

**Figure 1 sensors-24-02745-f001:**
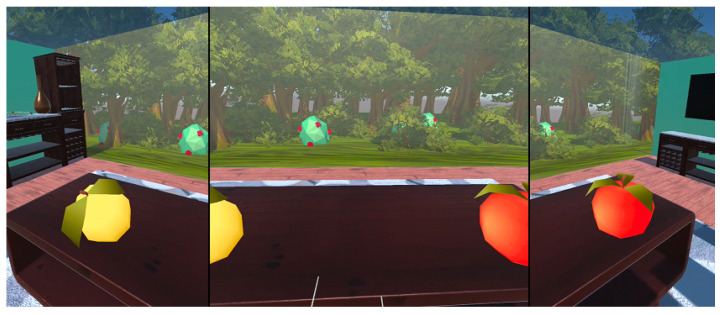
The view of the application in the beginning.

**Figure 2 sensors-24-02745-f002:**
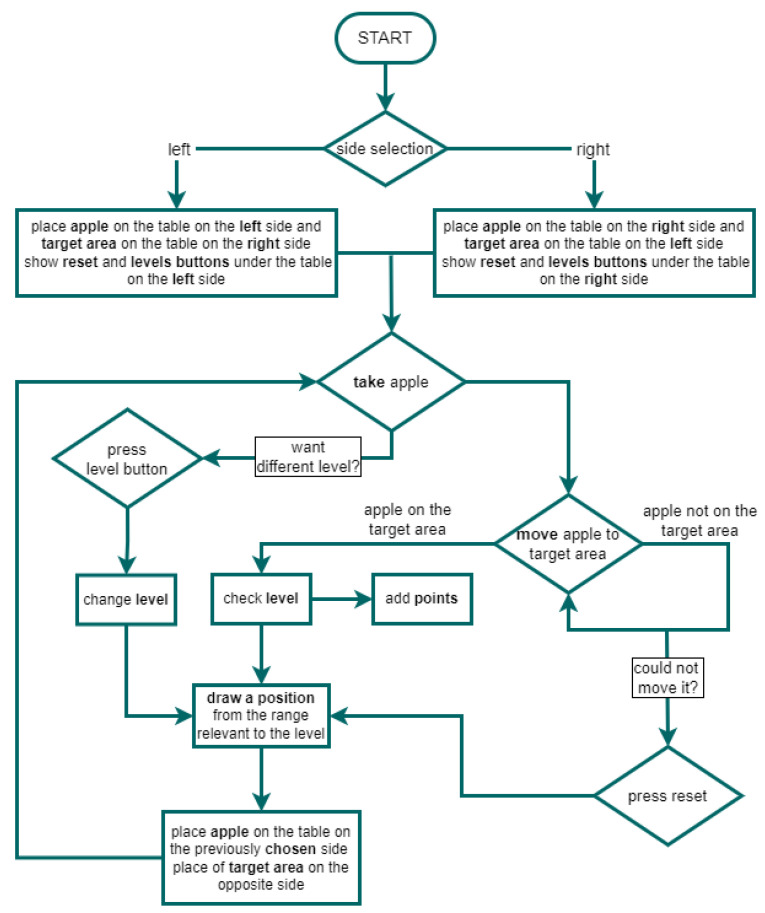
Functioning diagram of the application.

**Figure 3 sensors-24-02745-f003:**
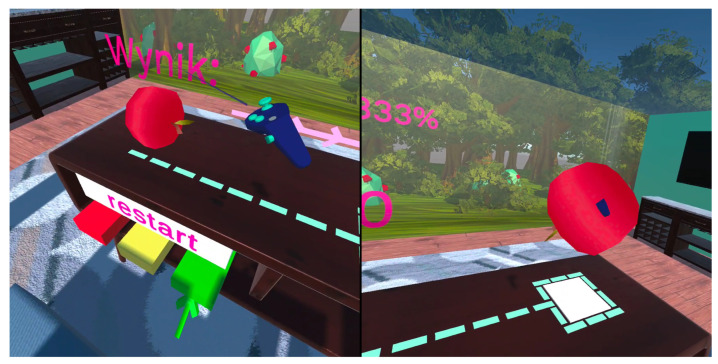
The game environment after selecting the left side.

**Figure 4 sensors-24-02745-f004:**
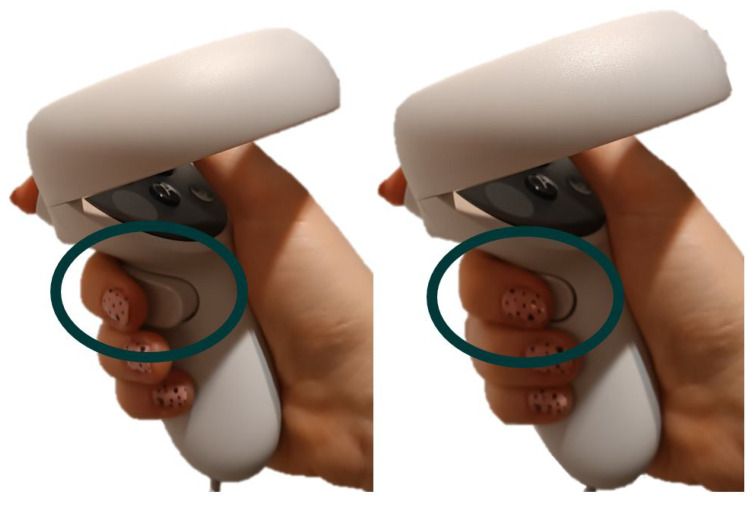
Meta Quest 2 controller used to operate in the game. Without the button pressed (**on the left**), with the button pressed (**on the right**).

**Figure 5 sensors-24-02745-f005:**
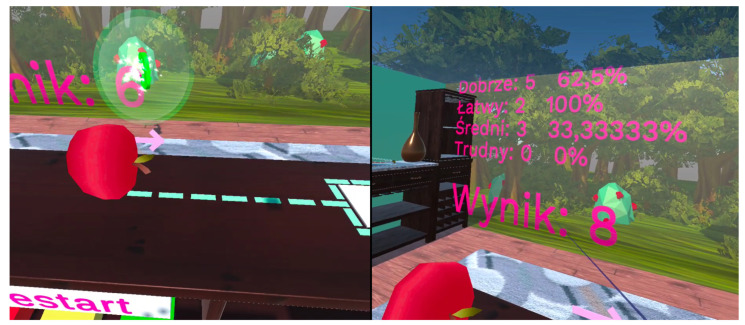
Motivational elements for patients (**on the left**) and statistics from the game (**on the right**).

**Table 1 sensors-24-02745-t001:** Participants groups in the study.

Group	Group Number	Number of Participants	Age	Gender
Control	Group 1	10	21.30, SD:0.48	3 F, 7 M
	Group 2	10	53.10, SD:7.26	7 F, 3 M
	Group 3	10	76.50, SD:8.53	4 F, 6 M
Study	Group 4	30	72.83, SD:8.14	11 F, 19 M

**Table 2 sensors-24-02745-t002:** Summary of using VR application among control groups.

Group Number	Age	Participant Was Convinced to Try VR [%]	Participant Can Do First Part of Exercise [%]	Participant Can Do Whole Exercise [%]	Participant Was Satisfied after Using VR Application [%]
Group 1	21.30, SD:0.48	100	100	100	100
Group 2	53.10, SD:7.26	90	100	90	90
Group 3	76.50, SD:8.53	50	80	90	90

**Table 3 sensors-24-02745-t003:** Summary of rehabilitation using VR with elderly post-stroke patients.

Category	Subcategory	Age	Patient Was Convinced to Try VR [%]	Patient Can Do First Part of Exercise [%]	Patient Can Do Whole Exercise [%]	Patient Was Moving during Exercise [%]	Patient Was Satisfied after VR Therapy [%]
All Patients		Mean 72.83, SD 8.14	46.67	80.00	56.67	83.33	73.33
Age range	60–74	Mean 67.33, SD 4.26	61.11	88.89	72.22	94.44	77.78
	75–90	Mean 81.08, SD 4.80	25.00	66.67	33.33	66.67	66.67
Stroke occurrence	<10 days	Mean 75.55, SD 9.70	27.27	54.55	27.27	81.82	63.64
	>10 days	Mean 71.26, SD 6.89	57.89	94.47	73.68	84.21	78.95
Patient’s condition	In good contact	Mean 70.41, SD 6.25	59.09	100.00	77.27	95.45	86.36
	In weak contact	Mean 79.50, SD 9.38	12.50	25.00	0.00	50.00	37.50

## Data Availability

Created game is available on: https://powerstudpw.itch.io/apples-game-for-stroke-rehabliti (accessed on 1 December 2023).

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
