# Peer review of "Study of the Possibility of Using Virtual Reality Application in Rehabilitation among Elderly Post-Stroke Patients"

_sensors, 2024, doi:10.3390/s24092745_

Round 1

Reviewer 1 Report (New Reviewer)

Comments and Suggestions for Authors

Dear authors,

your work about "Study of the Possibility of Using Virtual Reality Application in Rehabilitation Among Elderly Post-stroke Patients" has some importance for the literauture. Your study goal of "to test the feasibility of using virtual reality for the 107 rehabilitation of the upper limb among elderly people (over 60) after a stroke" in my oppinion is challeging to research and can help professionals in this area improve their capacities to deal with this specific population.

In general your work is well written and clear. For this reason i think your work is ready for publication.

congratulations

Author Response

Dear reviewer,
Thank you very much for your review and kind words about our work.

Reviewer 2 Report (Previous Reviewer 3)

Comments and Suggestions for Authors

The manuscript is the resubmitted version which I reviewed recently.

The text was slightly improved.

The main question addressed by the research: can the immersive virtual reality be used for the post-stroke rehabilitation among elderly people (over 60)?

Previous studies did not explore the use of virtual reality applications among such patients, the age of participants in previous studies was below 60.

Here the authors specifically studied the applicability of VR rehabilitation for the group of people over 60.

The authors recruited 60 participants, including young, mean-aged, and elderly control groups and the group of elderly post-stroke patients (60-89 years old).

Main conclusion of the paper is that people over 60 years of age, although sceptic at first contact with VR, are eventually satisfied with the therapy session and are interested in further therapy in this form. By using experimental approach, it was shown that indeed VR can be used for therapy among people over 60 years old. Unfortunately, not all patients can be convinced to try the technology for the first time.

The references of the paper are appropriate.

Author Response

Dear reviewer,
Thank you very much for your review and kind words about our work.

Reviewer 3 Report (Previous Reviewer 2)

Comments and Suggestions for Authors

The manuscript is already quite polished, with only one concern that needs to be addressed. The conclusion is overly lengthy and should be condensed into a single paragraph, focusing solely on the key message the author wants readers to take away.

Additional comment. Please see below.  

The main question addressed by the research is whether elderly post-stroke patients are able to use virtual reality (VR) applications in rehabilitation.  

The original and relevant part of this research is the investigation into the use of VR technology among elderly post-stroke patients, a population that is often excluded from technological advancements due to assumptions about their technological proficiency.

The paper addresses a gap in the field by focusing on the feasibility and effectiveness of VR rehabilitation for this specific group.  
Compared to other published material, this study adds evidence that elderly post-stroke patients can indeed benefit from VR rehabilitation, challenging the common perception that advanced technology is not suitable for this age group. It provides empirical data to support the integration of VR in post-stroke rehabilitation programs for patients of all ages.

  In terms of methodology, the authors could consider improving the study by including a larger sample size to increase the statistical power of the results. Additionally, they could control for variables such as prior experience with technology, level of education, and cognitive abilities, which may influence the patients' ability to use VR applications.  

The conclusions are consistent with the evidence and arguments presented. The study shows that elderly post-stroke patients are able to use VR applications, and their performance depends more on their mental and physical conditions rather than age. All main questions posed in the study were addressed through specific experiments, including self-assessment by patients, evaluation by physiotherapists, and patients' performance of exercises in VR.  

The references, tables and figures appear to be appropriate.

Overall, this study addresses an important gap in the field of post-stroke rehabilitation by investigating the use of VR technology among elderly patients. The findings are promising and suggest that VR can be a valuable tool in rehabilitation programs for patients of all ages.

End comments

Author Response

Dear reviewer,
Thank you very much for your review and kind words about our work. We have shortened the conclusion focusing on the key premise.

Thank you for your additional comments, especially for your comment on methodology. In this study we would not have been able to control for other variables such as experience with technology, education level, or cognitive ability for all patients. However, we will take this into consideration for our future work.

This manuscript is a resubmission of an earlier submission. The following is a list of the peer review reports and author responses from that submission.

Round 1

Reviewer 1 Report

Comments and Suggestions for Authors

The manuscript has been revised but it still does not fulfill general requrements to be a research article.

Author Response

Thank you very much for taking the time to review this manuscript. Unfortunately, due to the lack of comments made about the article, we are unable to address this review.
We would appreciate your comments  on what precisely we could improve to make our article better.

Reviewer 2 Report

Comments and Suggestions for Authors

This study presents an interesting approach to utilizing virtual reality (VR) in rehabilitation for elderly patients with upper limb paresis and unilateral spatial neglect (USN). However, there are several limitations and shortcomings that need to be addressed.

Firstly, the sample size of 60 individuals, while sufficient for a pilot study, may not be representative enough to generalize the results to a larger population. A larger sample size would provide more robust and reliable data, reducing the potential for bias and increasing the study's validity.

Secondly, the study only focuses on patients with specific neurological diseases, specifically upper limb paresis and USN. While this is a valid starting point, it would be beneficial to expand the study to include a more diverse range of neurological conditions to assess the wider applicability of VR in rehabilitation.

Moreover, the study only evaluates the ability of patients to use VR applications and their performance in VR exercises. It does not delve into the longer-term effects of VR-based rehabilitation, such as improvements in functional outcomes or reductions in disability. Future studies should aim to assess the long-term impact of VR rehabilitation on patient outcomes.

Additionally, the study mentions that the ability to correctly and fully perform an exercise in VR depends on several factors, including the ability to make logical contact. However, it does not provide a detailed analysis of these factors or offer specific strategies to address them. A more thorough investigation into the barriers to VR use by elderly patients and potential solutions would enhance the study's practical implications.

Finally, the study lacks a perfect control group, making it difficult to assess the relative effectiveness of VR-based rehabilitation compared to traditional rehabilitation methods. Future research should include one or more control groups to allow for a more rigorous evaluation of the benefits of VR in rehabilitation.

In summary, while this study provides valuable insights into the potential of VR in rehabilitation for elderly patients with neurological diseases, it suffers from limitations in sample size, disease diversity, outcome measures, factor analysis, and lack of a comparison group. Future studies should address these shortcomings to further develop and optimize VR-based rehabilitation programs for this patient population.

Reviewer 3 Report

Comments and Suggestions for Authors

The manuscript “Study of the Possibility of Using Virtual Reality Application in Rehabilitation Among Elderly Post-stroke Patients” reports novel data concerning the use of VR for post-stroke rehabilitation among people over 60. This subject is of great interest due to the increasing occurrence of stroke in the population and life expectance. The application was tested on a group of 60 individuals including 30 post-stroke patients. Study design seems appropriate, and conclusions are supported by the results.

Overall, the paper is nice, and the authors rightly note that “virtual reality offers many possibilities and is a promising tool in rehabilitation. The study gives hope that it can be used by people of all ages, including the elderly.”.

My only concern is that the paper is not related to the “Sensors” journal in any way according to aims and scope https://www.mdpi.com/journal/sensors/about.

Author Response

Thank you for your review. Regarding the appropriateness of publishing an article in the journal Sensors, we submitted an article to a Special Issue called "Extended Reality in Medicine and Healthcare: Methods, Technologies, Applications and Future Trends,". We believe that our work fits into this topic, particularly in the sections "XR technological innovations concerning head-mounted displays, tracking systems, and projectors" and "XR-enabled patient care, such as telemedicine, remote monitoring, and rehabilitation."

Reviewer 4 Report

Comments and Suggestions for Authors

The authors explore using virtual reality (VR) in post-stroke rehabilitation for elderly patients. They developed a VR application for upper limb paresis and tested it on a group including post-stroke patients with an average age of 72.83 years. Results indicate that neurological patients can use VR successfully, depending on factors like logical reasoning ability. Notably, age does not limit VR use; mental and physical conditions are crucial, suggesting VR's potential for post-stroke rehabilitation across all ages. The contributions of the work are fair and exciting and can be accepted after some revisions.

1. The introduction section should be updated with recent works on using VR/AR/MR in rehabilitation. Refer to this recent work summarizing all such state-of-the-art (https://doi.org/10.1007/978-981-16-9455-4_10). Thereafter, the author should highlight the novel contributions of the work.

2. The authors should provide more details about the virtual reality (VR) rehabilitation application and setup used in the study. Specifically, what features or elements were included in the VR environment to facilitate upper limb paresis and unilateral spatial neglect (USN) rehabilitation?

2. Did the authors use standardized assessment tools or metrics to measure improvements in patients' upper limb function and spatial neglect?

3. Regarding the demographic characteristics of your study population, what specific stroke level and type were represented among the participants? Did the authors observe any variations in the efficacy of the VR rehabilitation application based on the underlying different stroke conditions?

4. It is mentioned that the ability to correctly and fully perform exercises in VR depends on several factors, including the ability to make logical contact. The authors are suggested to elaborate on what "logical contact" means in the context of VR rehabilitation and how it influences patients' performance.

5. In the study, did the authors observe any correlations between patients' self-assessment of their VR experience and objective evaluations conducted by physiotherapists? How did these subjective and objective measures align with each other?

6. More insights into the technological aspects of the VR system are needed (as an extension of Comment 2). For example, what type of VR hardware (e.g., headsets, controllers) and software (e.g., interactive exercises, feedback mechanisms) were employed, and were there any specific design considerations for elderly users?

7. Considering the average age of your study participants (72.83 years), were there any challenges or limitations encountered in implementing VR technology within the rehabilitation setting for elderly individuals? How were these challenges addressed or mitigated during the study?

8. Moving forward, the authors should discuss the potential implications of your findings for the broader integration of VR technology in post-stroke rehabilitation settings.  The results should be compared with existing studies, quantitatively or qualitatively in a separate discussion section.
